# Time trend analysis of perinatal mortality, stillbirth, and early neonatal mortality of multiple pregnancies for each gestational week from the year 2000 to 2019: A population-based study in Japan

**Eijiro Hayata, Masahiko Nakata◉\*, Mineto Morita**

Department of Obstetrics and Gynecology, Toho University School of Medicine, Tokyo, Japan

\* masahiko.nakata@med.toho-u.ac.jp

## Abstract

Multiple pregnancies pose a high risk of morbidity and mortality in both mothers and infants; thus, obtaining reliable information based on a large population is essential to improve management. We used the maternal and child health statistics, which are published annually, from the database of the Ministry of Health, Labor, and Welfare. The data obtained were aggregated in 5-year intervals, and we used them to analyze the proportion of the number of births for each week of pregnancy to the total of each singleton and multiple pregnancy. For perinatal health indicators (perinatal mortality, stillbirth, and neonatal mortality), the obtained data were calculated and plotted on graphs for each week of pregnancy. Moreover, these indicators were calculated by dividing them into first twin and second twin fetuses. Stillbirth weights were aggregated in several groups, and a histogram was displayed. Between 2000 and 2019, there were 21,068,275 live births, 67,666 stillbirths, and 16,443 early neonatal deaths, excluding 7,148 (7,104 singletons, 44 multiple births) cases, in which the exact gestational weeks at birth were unknown. More than 95% of multiple pregnancies were twin births. Perinatal mortality, stillbirth, and early neonatal mortality rates in multiple pregnancies were the lowest at approximately 37 weeks of gestation and lower than those of single pregnancies at approximately 36 weeks of gestation. Perinatal mortality and stillbirth rates were higher during the delivery of the second twins than the first-born twins, but the early neonatal mortality rate remained approximately the same during the delivery of both twins. As the data in the government database are accumulated and published continuously, indicators can be calculated in the future using the method presented in this study. Further, our findings may be useful for policymaking related to managing multiple pregnancies.

## Introduction

Multiple pregnancies have a higher risk of maternal and fetal morbidity and mortality than singleton pregnancies. In particular, complicated multiple pregnancies force obstetricians to

**Data Availability Statement:** All relevant data are within the manuscript and its Supporting information files.

**Funding:** The authors received no specific funding for this work.

**Competing interests:** The authors have declared that no competing interests exist.

decide the specific timing of delivery, considering both maternal and fetal risks. Obstetricians are often required to predict the ideal timing of delivery, balancing between avoiding maternal complications that increase with later gestational age (e.g., gestational hypertension, pre-eclampsia [1, 2], placental abruption [2], thromboembolism [2]) and preventing fetal immaturity because of preterm birth. Moreover, diseases specific to multiple pregnancies, such as twin-to-twin transfusion syndrome, usually worsen the prognosis of newborns. In general, many studies have recommended setting the timing of delivery in multiple pregnancies earlier than that in a singleton pregnancy [3–6].

The latest population-based study conducted in Japan, which determined the appropriate timing of delivery in multiple pregnancies to reduce complications owing to fetal factors, dates back to 1996 [7]. Since then, no other large-scale data analyses of perinatal health indicators (perinatal mortality, stillbirth, and early neonatal mortality rate) have been reported.

Considering the constant changes in perinatal medicine, it is essential to update perinatal health indicators by gestational weeks and time of delivery, and determine the temporal changes of indicators to improve the management of multiple pregnancies in the future. In this study, we investigated the perinatal health indicators for each gestational week of multiple pregnancies that occurred between 2000 and 2019, based on the information available on the government database, and analyzed the time trends. Additionally, we analyzed the trends of indicators in singleton pregnancies and compared them to those in multiple pregnancies.

## Materials and methods

### Data source

We used the maternal and child health statistics, which are published annually, from the database of the Ministry of Health, Labour and Welfare (MHLW) [8]. In Japan, the "Family Register Act" stipulates that all births, deaths, and stillbirths that occur in Japan should be reported to MHLW. MHLW aggregates data based on the notification and publishes a database annually. The data collection rate is 100%, but it is not possible to grasp the extent of illegal births, stillbirths, or abortions. Live birth and stillbirth are grouped into one category for <20 gestational weeks and ≥49 weeks, respectively, and data for 20–48 weeks are published for each gestational week. Birth weights of <500 g and ≥6,900 g are grouped into one category, and data for 500–6,900 g are published for every 100 g. Births with defects are also included. As this national database is used as an official basis for policymaking, it is considered to be the most reliable database for population studies in Japan. The online database provides the number of live births, stillbirths, and early neonatal deaths that occur in each gestational week of multiple pregnancies since 2000 [9]. Using this information, we conducted a time-lapse analysis using data of 5-year-periods to evaluate perinatal health indicators associated with singleton and multiple pregnancies that occurred between 2000 and 2019 in Japan (S1–S4 Files).

### Perinatal health indicators

**Perinatal mortality rate.** Perinatal mortality rate is typically expressed as $1,000 \times$ (number of stillbirths after 22 weeks of gestation + number of early neonatal deaths) / (total number of births and stillbirths). Early neonatal death is defined as death within 1 week (7 days) of live birth.

The formula for perinatal mortality rate used in this study is as follows:

$$\text{Perinatal mortality rate at N weeks of gestation} = 1,000 \times (\text{number of stillbirths at N weeks of}$$
$$\text{gestation} + \text{the number of early neonatal deaths born at N weeks of gestation}) / (\text{number of}$$
$$\text{live births} + \text{stillbirths at N weeks of gestation}).$$

**Stillbirth rate.** In general, the stillbirth rate is expressed as 1,000 × (number of stillbirths) / (number of live births and stillbirths).

The formula for the stillbirth rate used in this study, based on the number of gestational weeks is as follows:

Stillbirth rate at N weeks of gestation $=$

1, 000 $\tilde{n}$ (the number of stillbirths at N weeks of gestation) /

(number of live births $+$ stillbirths at N weeks of gestation).

**Early neonatal mortality rate.** Typically, the early neonatal mortality rate is expressed as 1,000 × (number of early neonatal deaths) / (number of live births). In this study, the early neonatal mortality rate according to the number of gestational weeks was defined as follows:

Early neonatal mortality rate at N weeks of gestation $=$

1, 000 $\tilde{n}$ (number of early neonatal deaths at N weeks of gestation) /

(number of live births at N weeks of gestation).

These indicators were calculated in the "birth-based approach" [10]. The "fetuses-at-risk approach" is considered more appropriate as an epidemiological model of fetal mortality in a population with some maternal risk factors (e.g., smoking and hypertension). If multiple pregnancies are considered to be a risk factor for the fetuses, it is more appropriate to use the "fetuses-at-risk approach." However, in multiple pregnancies, specific issues should be considered. In cases where a fetus dies in the uterus, the gestational age at stillbirth of the deceased fetus is recorded as that at delivery of the other live infant. To use the "fetuses-at-risk approach," it is necessary to include the gestational age at the time when the fetus died in the uterus. However, especially in cases of dichorionic twins, the standard management in twins is continuing pregnancy until the term period, even if one fetus dies in the uterus. Therefore, it is usually impossible to estimate the exact intrauterine survival time of the dead fetus from the database. From these points of view, we thought that it would be better to apply the "birth-based approach" at this stage.

## Data analysis

The raw data obtained were aggregated every 5 years (2000–2004, 2005–2009, 2010–2014, and 2015–2019). We analyzed the proportion of the number of births for each week of pregnancy to the total of each singleton and multiple pregnancy. For perinatal health indicators (perinatal mortality, stillbirth, and neonatal mortality), the obtained data were plotted on graphs for each week of pregnancy without using a regression curve. Stillbirth weights were aggregated every 4 weeks until the extreme-moderate preterm (22–33 weeks), and late preterm (34–36 weeks) and term (after 37 weeks) were aggregated as one category each. A histogram was generated (S5 File).

## Ethical consideration

This study only used data that were aggregated and published nationwide and were freely available online; no human participants or animals were involved. In Japan, an institutional review board approval is not required for this type of study [11]. Consequently, the requirement for ethical approval for the current study was waived.

**Table 1. Time trends for multiple pregnancies and perinatal health indicators.**

| Year | 2000–2004 | 2005–2009 | 2010–2014 | 2015–2019 |
|---|---|---|---|---|
| **Live birth (total)** | 5,749,395 | 5,406,214 | 5,192,770 | 4,712,748 |
| **Stillbirth (total)** | 22,345 | 17,672 | 15,287 | 12,362 |
| **Early neonatal death (total)** | 5,772 | 4,345 | 3,520 | 2,806 |
| **Live birth (multiple pregnancy)** | 121,359 | 116,003 | 99,383 | 94,018 |
| Twins | 117,023 | 113,058 | 97,240 | 92,258 |
| Triplets or more | 4,336 | 2,945 | 2,143 | 1,760 |
| **Multiple births in total births (‰)** | 2.1 | 2.1 | 1.9 | 2.0 |
| **Twins in multiple births (‰)** | 96.4 | 97.5 | 97.8 | 98.1 |
| **Perinatal mortality rate (‰)** | | | | |
| Singleton | 5.0 | 4.2 | 3.7 | 3.3 |
| Multiple | 23.6 | 19.2 | 16.8 | 15.0 |
| **Stillbirth rate (%)** | | | | |
| Singleton | 4.0 | 3.3 | 3.0 | 2.7 |
| Multiple | 16.7 | 14.2 | 13.2 | 11.8 |
| **Early neonatal mortality rate (%)** | | | | |
| Singleton | 1.0 | 0.8 | 0.7 | 0.6 |
| Multiple | 6.9 | 5.0 | 3.5 | 3.2 |

## Results

### Time trends of births and perinatal health indicators for multiple pregnancies

Between 2000 and 2019, there were 21,068,275 live births, 67,666 stillbirths, and 16,443 early neonatal deaths, excluding 7,148 (7,104 singletons, 44 multiple births) cases, in which the number of gestational weeks at birth was <22 weeks or unknown. Table 1 shows the 5-year trends in the number, ratio, and perinatal health indicators of live births and multiple live births, respectively. There is a decreasing trend in the number of live births, which decreased by 18% (from 5,749,395 to 4,712,748) between 2000–2004 and 2015–2019. The number of multiple births also showed a decreasing trend, which decreased by 22% (from 121,359 to 94,018) between 2000–2004 and 2015–2019. The ratio of multiple pregnancy live births to the total live births remained unchanged at approximately 2%. The ratio of twin births to the total number of multiple births showed an increasing trend from 96.4% to 98.1%. Moreover, all perinatal health indicators showed decreasing trends in both singleton and multiple births.

### Time trends for the distribution of singleton and multiple pregnancies in each gestational week

Fig 1 shows the 5-year trend in the distribution of singleton and multiple pregnancies over each N gestational week. The highest proportion of gestational weeks in singleton pregnancies was 39 weeks, which has remained unchanged over the past 20 years. The number of deliveries at 38 weeks of gestation increased, and the proportion of deliveries after 40 weeks of gestation decreased. Conversely, the highest proportion of gestational weeks in multiple pregnancies was 37 weeks, which showed an increasing trend in the past 20 years. The next highest proportion of gestational weeks was 36 weeks, followed by 35 weeks. The proportion of deliveries after 38 weeks decreased.

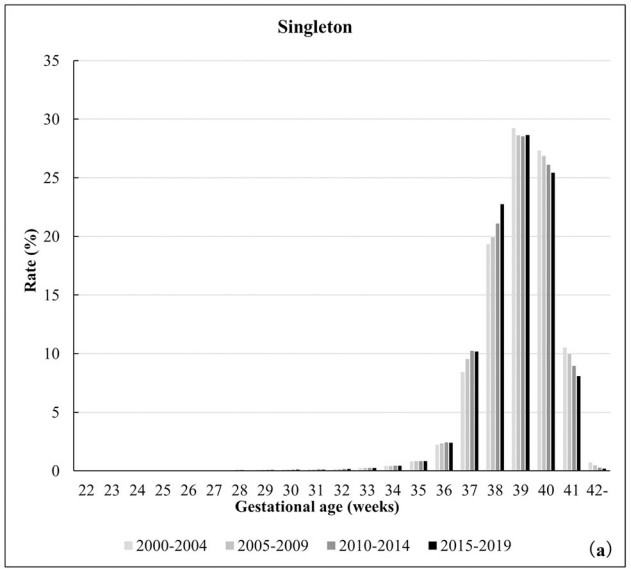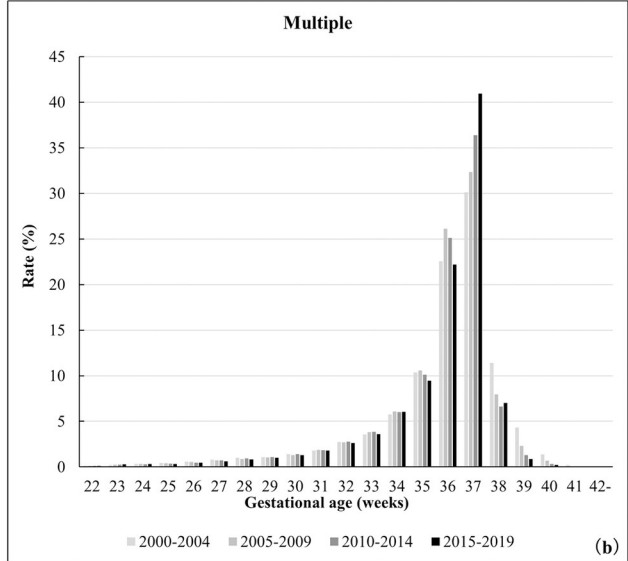

**Fig 1. Time trends for the distribution of singleton and multiple pregnancies based on the gestational age in weeks.** (a) Singleton, (b) Multiple.

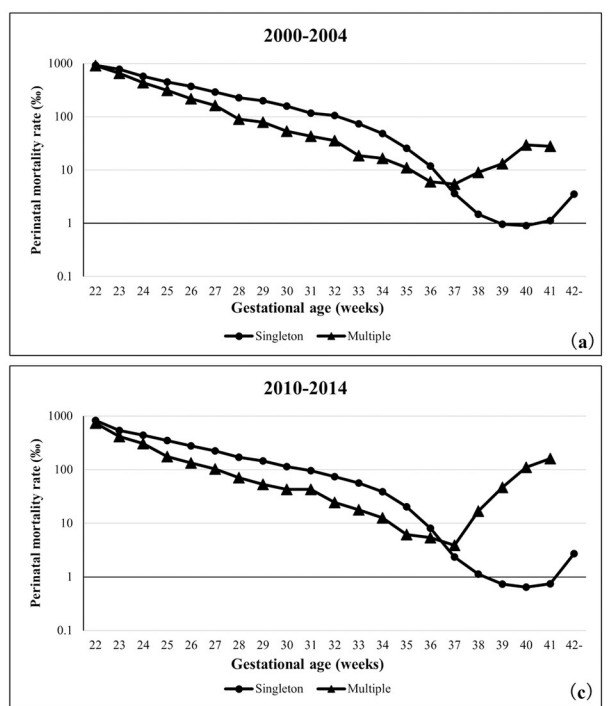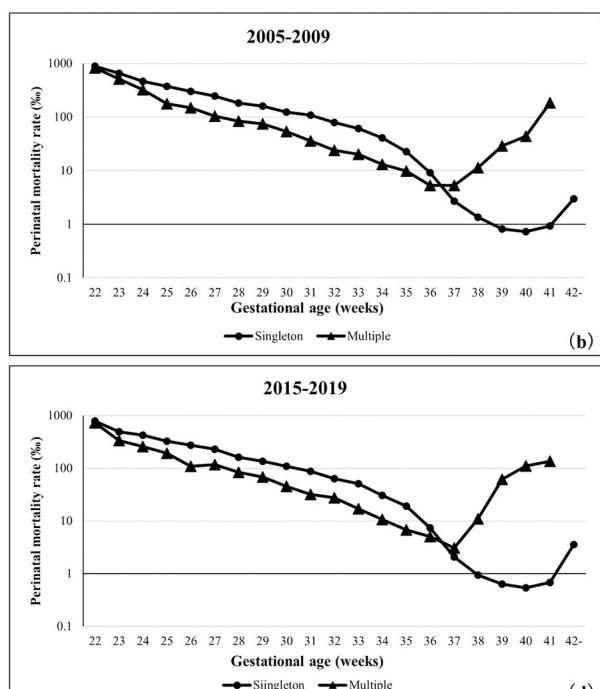

**Fig 2. Perinatal mortality rates of singleton and multiple pregnancies in each gestational week.** (a) 2000–2004 years, (b) 2005–2009 years, (c) 2010–2014 years, (d) 2015–2019 years.

### Time trends for the perinatal mortality rate based on the gestational age

Fig 2 shows the 5-year trends of perinatal mortality rates of singleton and multiple pregnancies based on the gestational age. The same trend was observed in each 5-year-period. The perinatal mortality rates of both singleton and multiple pregnancies showed a decreasing trend between 22 and 37 weeks of gestation. In singleton pregnancies, the perinatal mortality rate was the lowest between 39 and 41 weeks of gestation, while in multiple pregnancies, the perinatal mortality rate showed an increasing trend after 38 weeks of gestation. However, the perinatal mortality rate was lower in multiple pregnancies between 22 and 36 weeks of gestation than in singleton pregnancies.

### Time trends for stillbirth rate based on the gestational age

Fig 3 shows the 5-year trends of stillbirth rates of singleton and multiple pregnancies based on the gestational age. The same trend was observed in each 5-year-period. The stillbirth rates of both singleton and multiple pregnancies showed a decreasing trend between 22 and 37 weeks of gestation. The stillbirth rate in singleton pregnancies was the lowest between 39 and 41 weeks of gestation, while in multiple pregnancies, the stillbirth rate increased after 38 weeks of gestation. Additionally, the stillbirth rate was lower in multiple pregnancies between 22 and 36 weeks of gestation than in singleton pregnancies.

### Time trends for early neonatal mortality rate based on the gestational age

Fig 4 demonstrates the 5-year trends of early neonatal mortality rates of singleton and multiple pregnancies based on the number of gestational weeks. The early neonatal mortality rates of

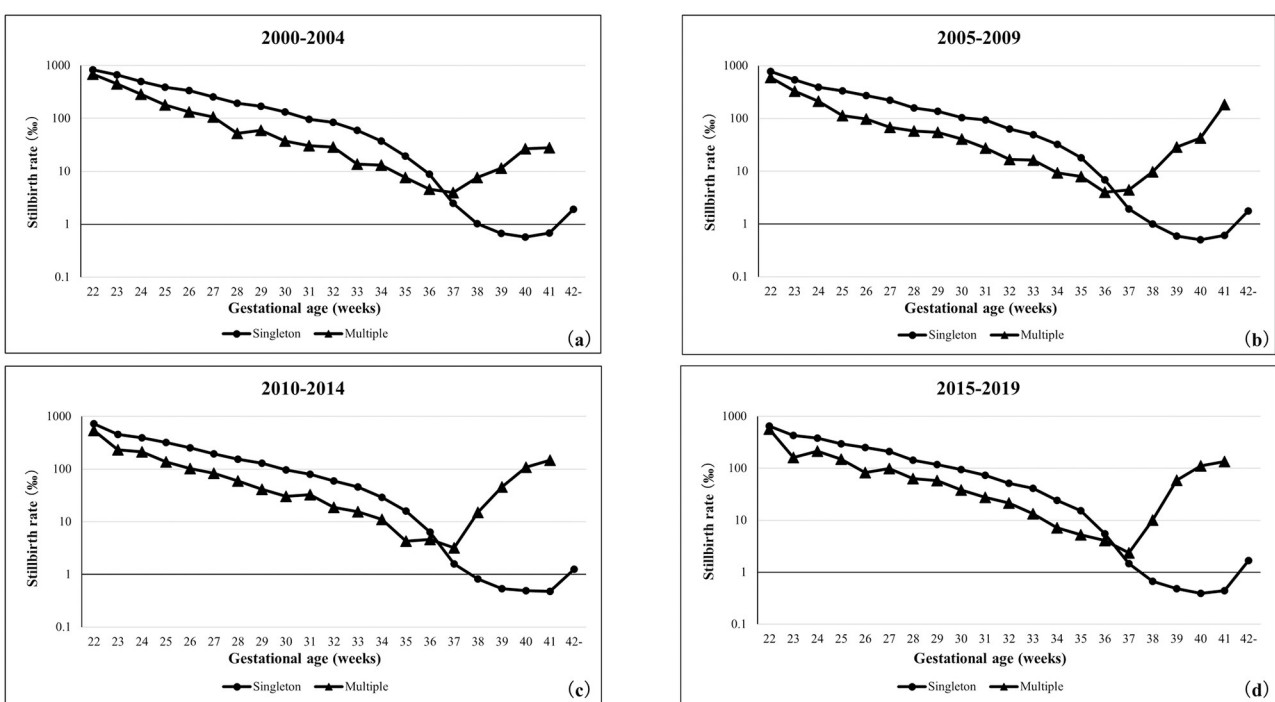

**Fig 3. Stillbirth rates of singleton and multiple pregnancies in each gestational week.** (a) 2000–2004 years, (b) 2005–2009 years, (c) 2010–2014 years, (d) 2015–2019 years.

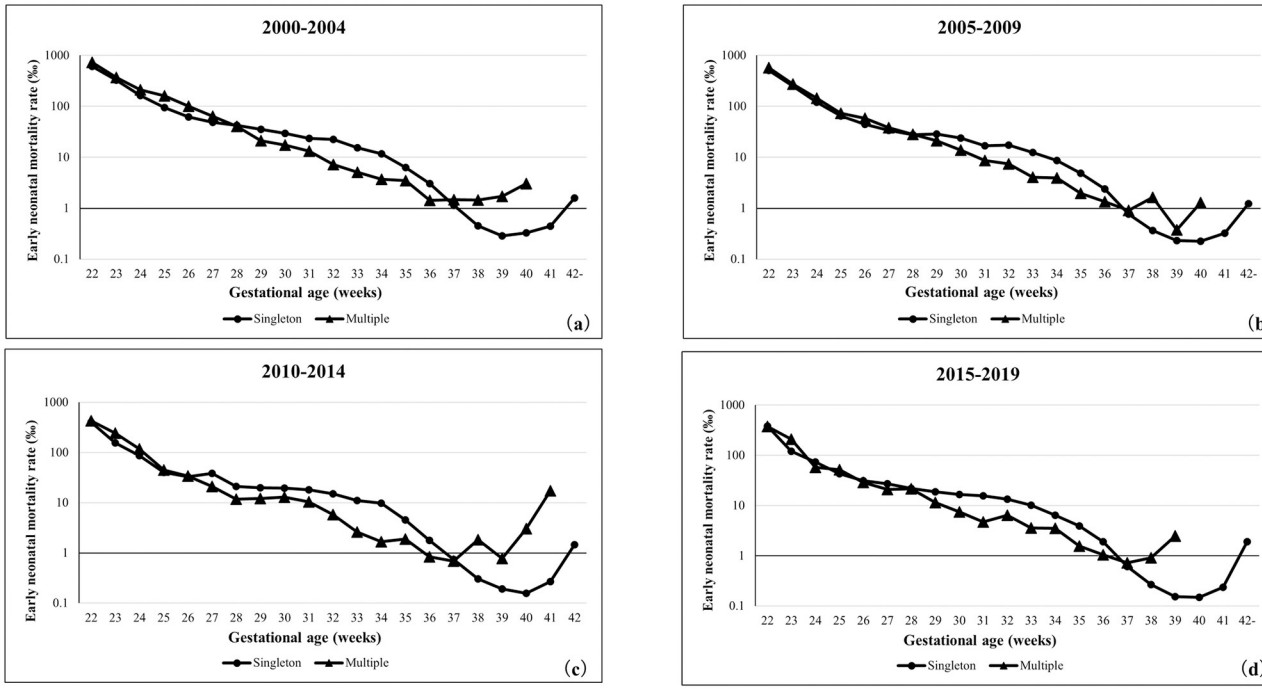

**Fig 4. Early neonatal mortality rate in each gestational week of singleton and multiple pregnancies.** (a) 2000–2004 years, (b) 2005–2009 years, (c) 2010–2014 years, (d) 2015–2019 years.

both singleton and multiple pregnancies showed a decreasing trend between 22 and 37 weeks of gestation. In singleton pregnancies, the early neonatal mortality rate was lowest between 39 and 41 weeks of gestation, while in multiple pregnancies, the early neonatal mortality rate decreased after 38 weeks of gestation. Furthermore, the early neonatal mortality rate was similar between singleton and multiple pregnancies up to approximately 28 weeks of gestation and slightly lower in multiple pregnancies between 28 and 36 weeks of gestation.

## Comparison of the perinatal outcomes of the first- and second-born twins

Fig 5 shows the trends of perinatal mortality, stillbirth, and early neonatal mortality rates of twins by birth order. (S6 File) The perinatal mortality and stillbirth rates were approximately twice as high in the second-born twins compared with the first-born twins, while the difference in early neonatal mortality rates was negligible between the twins. Fig 6 shows the birth weight trends of stillborn infants born after 22 weeks of gestation. After 30 weeks of gestation, stillborn infants weighing <500 g were the most frequently observed.

## Discussion

In this study, the perinatal health indicators of multiple pregnancies in each gestational week was calculated using data published by the government. First, the rate of multiple pregnancies has remained unchanged at approximately 2% in the past 20 years. The perinatal mortality, stillbirth, and early neonatal mortality rates were found to be the lowest at approximately 37 weeks of gestation in multiple pregnancies, and these trends were unchanged through 2000–2019. Second, the perinatal mortality and stillbirth rates were higher during the delivery of the

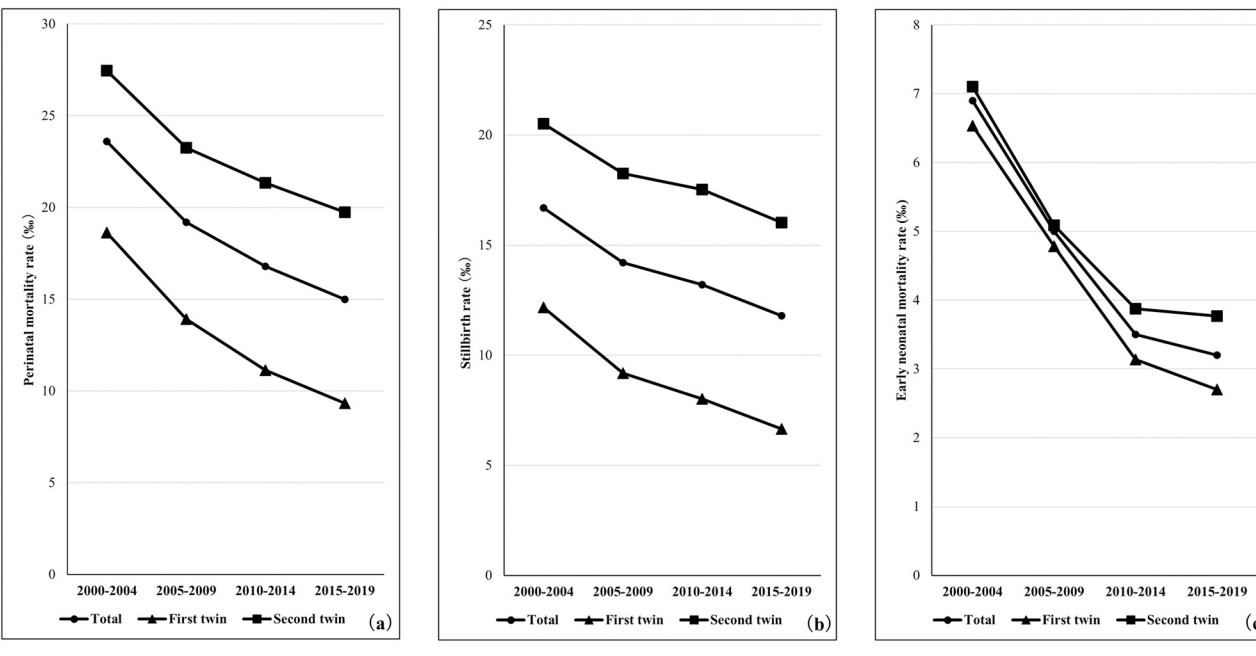

**Fig 5. Time trends in perinatal mortality, stillbirth, and early neonatal mortality rates of first and second twins.** (a) Perinatal mortality rate, (b) Stillbirth rate, (c) Early neonatal mortality rate.

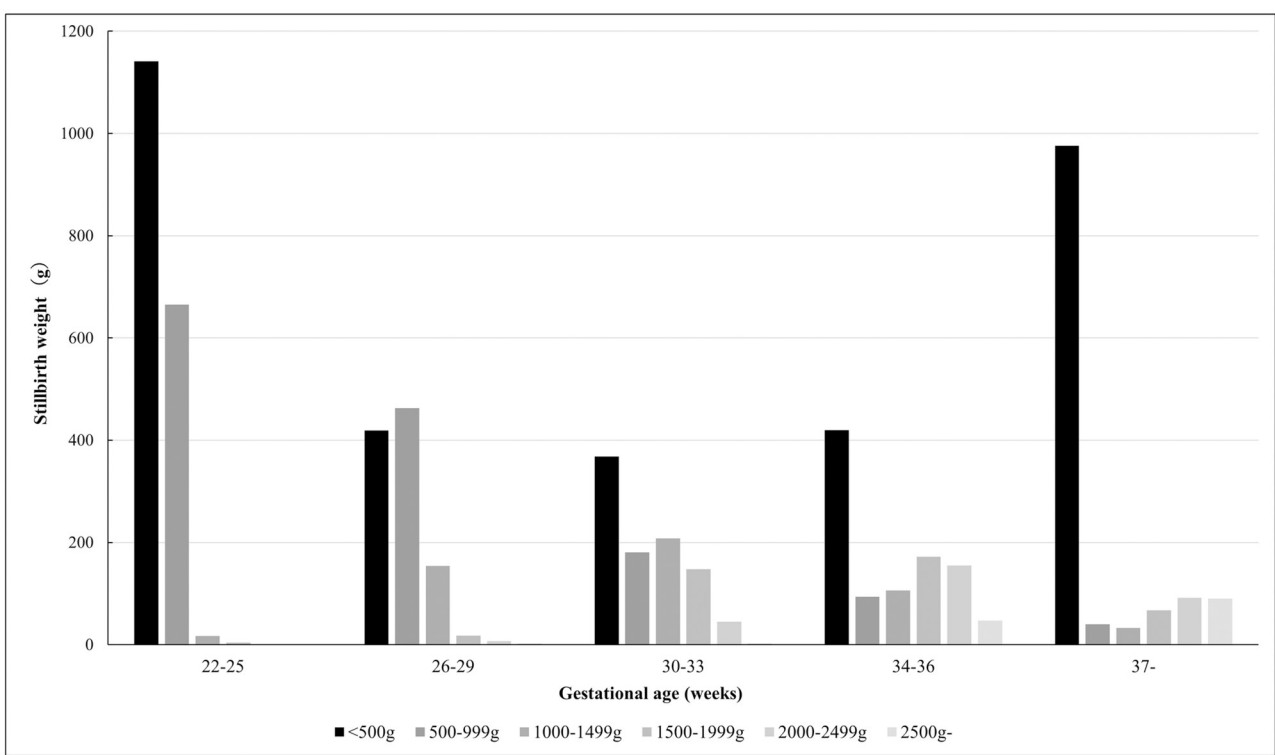

**Fig 6. Birth weight time trends of stillbirth infants.**

second twins than the first-born twins, but the early neonatal mortality rate remained approximately the same in the delivery of both twins.

The strength of this study is that it is suitable for providing a temporal overview of the perinatal health indicators in Japan, as we have used the government database, which is considered reliable, albeit with a few exceptions. Furthermore, as the data will be accumulated and published in the future, the method used in this study can be used to calculate these indicators in the years to come, which also allows for the comparison of these indicators with other countries.

The limitations of this study are as follows: (1) we were unable to examine the background factors of the mothers and fetuses, gestational complications, and differences in perinatal health indicators caused by chorionicity and amnionicity during multiple pregnancies, and (2) we were unable to analyze the differences in perinatal outcomes of different modes of delivery, as well as the long-term prognoses of the newborns. Hence, the optimal timing of delivery for multiple pregnancies should not be determined by using this study's results alone. Our results should be used only as a reference.

## Trends in births and the perinatal health indicators due to multiple pregnancy

In Japan, the rate of multiple pregnancies has remained unchanged at approximately 2% in the past 20 years. This rate, which varies widely by race and nationality, was reported to be 0.3–1.0% worldwide [12–14]. From the 1980s to 2010s, the rate of multiple pregnancies increased, especially in developed countries [12–14]. It was also reported that the multiple pregnancy rate is particularly high in urban areas within the same country [15]. The increase in the number of multiple pregnancies has been attributed to two main factors: (1) the natural increase because of the increase in advanced maternal age pregnancies and (2) the artificial increase owing to the increased use of assisted reproduction technologies [16]. In the United States of America, the multiple pregnancy rate has been declining again since the 2010s [17, 18]. In Japan, the number of transferable embryos has been strictly limited by the guidelines of the Japanese Society for Reproductive Medicine in 2007, when the use of assisted reproduction technologies accounted for 1.8% of all newborns. This restriction is thought to be one of the reasons why an increase in multiple pregnancies has been prevented [19].

In this study, the perinatal mortality, stillbirth, and early neonatal mortality rates were found to be the lowest at approximately 37 weeks of gestation in multiple pregnancies. A large cohort study that was conducted in 2020 [20] reported that the short- and long-term prognoses of newborn twins were most favorable at approximately 37 weeks of gestation, which is consistent with the guidelines provided by the United States of America, United Kingdom, and Japan [3, 4, 6]. A randomized clinical trial was conducted to investigate the ideal timing of delivery of twin pregnancies, but was discontinued because of the insufficient sample size [21]. Moreover, because of the difficulty of conducting randomized control trials due to ethical concerns, we believe that population-based studies, such as the present study, will continue to be useful in the future.

In this study, the perinatal mortality, stillbirth, and early neonatal mortality rates in multiple pregnancies were lower than those in singleton pregnancies up to approximately 36 weeks of gestation. These indicators in all multiple pregnancies were approximately five times higher than those in singleton pregnancies; however, the short-term prognoses of preterm newborns were better in multiple pregnancies. Similar trends were observed in the United States of America, where the neonatal mortality rate of low-birth-weight newborns was reported to be lower in multiple than in singleton pregnancies [13]. One reason for these paradoxical mortality curves (Figs 2–4) in this study may be the fact that this study was conducted using the

"birth-based approach" [10]. It has been argued this paradoxical appearance of gestational age-specific perinatal mortality curves in twins vs. singletons [22–24]. Yudkin et al. tried to explain this paradox and proposed the "fetuses-at-risk approach" formulation [25]. To use the aforementioned approach, it is necessary to include the gestational age at the time when the fetus died in the uterus. If the estimated period of death of the deceased twin fetus can be tracked on the database, it will be possible to apply the "fetuses-at-risk approach" to verify this paradox.

Another possible reason for these paradoxical mortality curves is assumed to be the fact that multiple pregnancies are often managed at large medical institutions consisting of numerous medical staff and advanced medical equipment, mainly in perinatal, maternal, and child medical centers in Japan. Approximately 50% of deliveries in Japan are handled in obstetric clinics, most of which are low-risk pregnancies. In contrast, approximately 80% of high-risk multiple pregnancies are managed in large hospitals, such as perinatal centers [26]. Therefore, early access to medical care during sudden obstetric complications, emergency illnesses and delivery of preterm infants may be one of the reasons why these indicators are low in multiple pregancies. For example, from 2015 to 2019, 79.7% (73,585/92,258) of children born from multiple pregnancies are occupied by the perinatal registration facility of the Japan Society of Obstetrics and Gynecology (JSOG), and the same tendency is observed in other age groups [26]. As all the JSOG-registered perinatal facilities (408 facilities in 2019) [26] belong to large hospitals, including perinatal medical centers, it can be assumed that the majority of multiple pregnancies were managed as high-risk pregnancies, which may have contributed to the improvement in perinatal outcomes of preterm newborns.

### Perinatal outcome of the first and second twins

Perinatal mortality and stillbirth rates were higher during the delivery of the second-born twins than the first-born twins, but the early neonatal mortality rate remained approximately the same in the delivery of both twins. Some studies have reported that deliveries of the second twins had a higher perinatal mortality rate than those of the first-born twins [27, 28]; however, they did not report the discrepancy between the perinatal mortality rates of the two deliveries, which was found to be twice as much in our study. However, there was a little-to-no discrepancy in the outcomes of newborns (i.e., the early neonatal mortality rates) between singleton and multiple pregnancies. The majority of stillborn infants delivered after 34 full weeks of gestation had a birth weight of <500 g (Fig 6). In view of this, in some cases, one of the fetuses might have died before 22 weeks in the uterus, which might have been recorded as stillbirth during the delivery of both twins after 34 weeks of gestation. Such circumstances could have caused a discrepancy between stillbirth and perinatal mortality rates. In this study, the early neonatal mortality rate of second-born twins was 1.4 times higher than that of the first-born twins (3.8 vs. 2.7 per 1,000), which is close to the figures reported by a study in the United States of America [27]. Therefore, the differences in perinatal mortality and stillbirth rates between the two twins may not indicate the actual status of perinatal care in Japan.

### Conclusions

In this study, the perinatal mortality, stillbirth, and early neonatal mortality rates were the lowest at approximately 37 weeks of gestation in multiple pregnancies. In twin pregnancies, the values of the perinatal health indicators were lower than those in singleton pregnancies until 36 weeks of gestation. As the analysis was conducted using data from the government database, this study is capable of overviewing the trends in the perinatal health indicators for the entire population of Japan. Furthermore, since these government data will be contiuously accumulated and published in the future, trends in these indicators can be demonstrated using the

method that we presented in this study. We believe that the findings of this study may be useful as reference information for developing health policies related to the management of multiple pregnancies, which also allows for comparison of these indicators with other countries.

## Supporting information

**S1 File. Live birth_multiple pregnancy.**
(XLSX)

**S2 File. Distribution of gestational weeks of delivery.**
(XLSX)

**S3 File. Indicators_singleton pregnancy.**
(XLSX)

**S4 File. Indicators_multiple pregnancy.**
(XLSX)

**S5 File. Birth weight of stillbirth_multiple pregnancy.**
(XLSX)

**S6 File. Indicatotrs between first and second twins.**
(XLSX)

## Author Contributions

**Conceptualization:** Eijiro Hayata.

**Data curation:** Eijiro Hayata.

**Formal analysis:** Eijiro Hayata, Masahiko Nakata.

**Methodology:** Eijiro Hayata.

**Supervision:** Masahiko Nakata, Mineto Morita.

**Writing – original draft:** Eijiro Hayata.

**Writing – review & editing:** Masahiko Nakata, Mineto Morita.

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
