## [Decision Letter · Decision Letter 0]

7 Mar 2022

PONE-D-21-26832Time trend analysis of maternal and child health indicators of multiple pregnancies from year 2000 to 2019 for each gestational week: A population-based study in JapanPLOS ONE

Dear Dr. Nakata,

Thank you for submitting your manuscript to PLOS ONE. After careful consideration, we feel that it has merit but does not fully meet PLOS ONE’s publication criteria as it currently stands. Therefore, we invite you to submit a revised version of the manuscript that addresses the points raised during the review process. Please submit your revised manuscript by March 6, 2022. If you will need more time than this to complete your revisions, please reply to this message or contact the journal office at plosone@plos.org. Please include the following items when submitting your revised manuscript:A rebuttal letter that responds to each point raised by the academic editor and reviewer(s). You should upload this letter as a separate file labeled 'Response to Reviewers'.A marked-up copy of your manuscript that highlights changes made to the original version. You should upload this as a separate file labeled 'Revised Manuscript with Track Changes'.An unmarked version of your revised paper without tracked changes. You should upload this as a separate file labeled 'Manuscript'.

We look forward to receiving your revised manuscript.

Kind regards,

Alireza Abdollah Shamshirsaz

Academic Editor

PLOS ONE

2. PLOS requires an ORCID iD for the corresponding author in Editorial Manager on papers submitted after December 6th, 2016. Please ensure that you have an ORCID iD and that it is validated in Editorial Manager. To do this, go to ‘Update my Information’ (in the upper left-hand corner of the main menu), and click on the Fetch/Validate link next to the ORCID field. This will take you to the ORCID site and allow you to create a new iD or authenticate a pre-existing iD in Editorial Manager. Please see the following video for instructions on linking an ORCID iD to your Editorial Manager account: https://www.youtube.com/watch?v=_xcclfuvtxQ.

Thanks a lot for submitting your manuscript to PLOS ONE. I am looking forward to seeing your final manuscript soon. 

Reviewers' comments:

Reviewer's Responses to Questions

**Comments to the Author**

1. Is the manuscript technically sound, and do the data support the conclusions?

Reviewer #1: Yes

Reviewer #2: Partly

2. Has the statistical analysis been performed appropriately and rigorously? 

Reviewer #1: No

Reviewer #2: No

3. Have the authors made all data underlying the findings in their manuscript fully available?

Reviewer #1: No

Reviewer #2: Yes

4. Is the manuscript presented in an intelligible fashion and written in standard English?

Reviewer #1: Yes

Reviewer #2: Yes

5. Review Comments to the Author

Reviewer #1: I read the manuscript entitled "Time trend analysis of maternal and child health indicators of multiple pregnancies

from year 2000 to 2019 for each gestational week: A population-based study in Japan" with great interest, there are, however, some concerns as follow:

Comment 1.

Lines 23-24. Since more than 95% of multiple pregnancies lead to twin births, the indicators obtained were considered to be representative of twin pregnancies.

This statement is not generalizable to all multiple pregnancies please rephrase it.

Comment 2.

Methodology of abstract is not clear or even in some parts is missing.

Comment 3.

Lines 27-28. This may be because most multiple pregnancies were managed in large-scale medical institutions.

Based on your results you cannot conclude this.

Comment 4.

Line 75. Authors have evaluated perinatal, stillbirth and early neonatal mortality. How did they cover maternal health index?

Comment 5.

There is no information on how time laps analysis was done.

Comment 6.

Start the discussion with your main findings.

Reviewer #2: PONE-D-21-26832

This population-based study examines stillbirth, neonatal and perinatal mortality rates among multiple births in Japan over the period 2000-2019. The study also examines perinatal mortality rates between first and second twin in twin births, as well as birthweight distribution by gestational age among stillbirths. The manuscript can be further strengthened by addressing the following comments.

Major Comments

1. The authors have used a ‘births-based’ approach to calculate gestational-age specific stillbirth and perinatal mortality rates, that is, the denominator equals total births (live births + stillbirths) at the specific gestational age. The authors should provide a strong justification for why this approach was used rather than the ‘fetus-at-risk’ approach. Use of the ‘births-based’ approach results in intersecting gestational-age specific mortality curves (such as those seen in Figures 2-4) in which the ’higher risk’ group appear to have lower mortality rates at earlier gestations compared to the ’lower risk’ group. It has been argued that the appropriate denominator should be those at risk of stillbirth which is delivered and undelivered fetuses surviving to the specific gestational age rather than births at the specific gestational age.

Refs:

Yudkin 1987 Lancet 1987;1:1192-4 https://doi.org/10.1016/S0140-6736(87)92154-4

Joseph & Kramer Acta Obstetricia et Gynecologica Scandinavica 2018 doi.org/10.1111/aogs.13194

Smith Am J Obstet Gynecol . 2005 Jan;192(1):17-22. doi: 10.1016/j.ajog.2004.08.014.

Other Comments

Title

2. Maternal and child health indicators is a very broad term and as this manuscript focuses exclusively on perinatal mortality, the authors should consider using more specific terms in the manuscript title.

Abstract

3. The authors should include a brief description of the study population, state the statistical methods used to analyse the data, and present some results within the abstract. For example, where there any major exclusion criteria? What was the gestational age threshold for the database? The authors should also consider presenting overall mortality rates before describing trends over time.

4. Lines 27-34 should be omitted. These are explanations which are more suitable for the discussion section than the abstract.

Introduction

5. The second sentence (lines 42-46) should be revised to improve clarity.

6. Please also include references for lines 42-46

Methods

7. For international audiences who are not familiar with the Ministry of Health, Labour and Welfare database, the authors should provide additional information about the database and the quality of the data collected. For example, what proportion of the birthing population in Japan is captured by the database? What are the criteria for inclusion in the database – what gestational age or birthweight thresholds are used? Are births with a congenital anomaly included or excluded? How valid and reliable are the data from the database? Are there any published validation studies using these data?

8. The authors should state which statistical methods were used. Have the authors considered formal tests such as linear regression for the trends over time presented?

Results

9. In the text of the results, the authors should first describe the study population before referring to Table 1. For example, present the total number of births over the study period, describe any births excluded from the analyses for whatever reasons.

10. In Table 1, please include the number of stillbirths and neonatal deaths

11. Mortality rates should be presented as per 1000 rather than per 100 (%).

12. Lines 124-128 – Perhaps this is referring to the highest proportion of births occurring at 39 weeks for singletons and 37 weeks for multiples, rather than maximum number of gestational weeks?

Discussion

13. The study limitations should be moved so that they follow on from the discussion of study strengths.

14. If the ‘fetus-at-risk’ approach is used, the discussion on intersecting mortality curves (lines 227-243) will likely need to be revised.

Reviewer #1: **Yes: **Kamran Hessami

Reviewer #2: No

---

## [Author Response · Author response to Decision Letter 0]

14 Apr 2022

Responses to journal requirements

We would like to thank you for your critical comments and insightful suggestions, which have helped us improve our manuscript. As indicated in the responses below, we have integrated all your comments and suggestions in the revised version of our manuscript.

[Comment]

[Response]

We have reedited the manuscript format according to the style stipulated by PLOS ONE.

[Comment]

2. PLOS requires an ORCID iD for the corresponding author in Editorial Manager on papers submitted after December 6th, 2016. Please ensure that you have an ORCID iD and that it is validated in Editorial Manager.

[Response]

We have registered the ORCID iD of the corresponding author in the Editorial Manager.

The ID is: Masahiko Nakata 0000-0002-3621-1251

[Comment]

3. Please include captions for your Supporting Information files at the end of your manuscript, and update any in-text citations to match accordingly.

[Response]

We have added the supporting information with captions at the end of the manuscript.

Additionally, we have updated the in-text citations referring to the supporting information in the text.

Responses to Reviewer 1

We would like to thank you for your critical comments and insightful suggestions, which have helped us improve our manuscript. As indicated in the responses below, we have integrated all your comments and suggestions in the revised version of our manuscript.

[Comment]

1. Lines 23-24. Since more than 95% of multiple pregnancies lead to twin births, the indicators obtained were considered to be representative of twin pregnancies. This statement is not generalizable to all multiple pregnancies please rephrase it.

[Response]

As the reviewer pointed out, generalizing the data obtained in this study as “twin pregnancy data” is inaccurate. Therefore, in this paper, the terms are unified as “multiple pregnancies.”

[Comment]

2. Methodology of abstract is not clear or even in some parts is missing.

[Response]

Following the reviewer's advice, we have added a brief description of our methodology to the abstract. (Lines: 21–28)

[Comment]

3. Lines 27-28. This may be because most multiple pregnancies were managed in large-scale medical institutions. Based on your results you cannot conclude this.

[Response]

We acknowledge that the description in the Discussion section was insufficient. Approximately 50% of all deliveries in Japan are handled in obstetric clinics, most of which are low-risk pregnancies. On the other hand, about 80% of high-risk multiple pregnancies are managed in large hospitals such as perinatal centers. Therefore, early access to sudden obstetric complications, emergency illnesses and preterm infants may be one of the reasons why these indicators result in low values.

This point has been described in discussion (Lines: 297–309) and the concerned information from the abstract has been deleted.

[Comment]

4. Line 75. Authors have evaluated perinatal, stillbirth and early neonatal mortality. How did they cover maternal health index?

[Response]

The term we used was inaccurate. Therefore, the term "maternal and child health indicators" in original manuscript has been changed to "perinatal related health indicators (perinatal, stillbirth, and neonatal mortality)" in the revised manuscript.

[Comment]

5. There is no information on how time laps analysis was done.

[Response]

The raw data obtained were aggregated for every 5 years (2000–2004, 2005–2009, 2010–2014, and 2015–2019). We analyzed the proportion of the number of births for each week of pregnancy to the total for each singleton and multiple pregnancy. For perinatal related health indicators (perinatal mortality, stillbirth, and neonatal mortality), the obtained data were plotted on graphs for each week of pregnancy without using a regression curve. Stillbirth weights were aggregated every 4 weeks, with the exception of extreme-moderate preterm (22–33 weeks), late preterm (34–36 weeks) and term (after 37 weeks), which were aggregated as one category each. The stillbirth weights were displayed as a histogram. 

We have added a "data analysis" subsection in "materials and methods" section. (Lines: 117–125)

[Comment]

6. Start the discussion with your main findings.

[Response]

In accordance with your suggestion, we have started the Discussions with our main findings.

We would like to again thank you for your constructive comments, which have highly enriched our manuscript.

Responses to Reviewer 2

We would like to thank you for your critical comments and insightful suggestions, which have helped us improve our manuscript. As indicated in the responses below, we have integrated all your comments and suggestions in the revised version of our manuscript.

[Comment]

1. The authors have used a ‘births-based’ approach to calculate gestational-age specific stillbirth and perinatal mortality rates, that is, the denominator equals total births (live births + stillbirths) at the specific gestational age. The authors should provide a strong justification for why this approach was used rather than the ‘fetus-at-risk’ approach. Use of the ‘births-based’ approach results in intersecting gestational-age specific mortality curves (such as those seen in Figures 2-4) in which the ’higher risk’ group appear to have lower mortality rates at earlier gestations compared to the ’lower risk’ group. It has been argued that the appropriate denominator should be those at risk of stillbirth which is delivered and undelivered fetuses surviving to the specific gestational age rather than births at the specific gestational age.

[Response]

We thank the reviewer for their thought-provoking advice.

As the reviewer pointed out, the paradoxical mortality curves in this study can be explained by the point that this study was conducted using the "birth-based approach." The "fetuses-at-risk approach" is considered more appropriate as an epidemiological model of fetal mortality in a population with some maternal risk factors (e.g., smoking, hypertension). If multiple pregnancies are considered to be a risk factor for the fetuses, it is more appropriate to use the "fetuses-at-risk approach." However, in the case of multiple pregnancies, specific issues should be considered. In the case where one fetus died in utero, the gestational age at stillbirth of the deceased fetus is recorded as the gestational age at delivery with the other live infant. To use the reliable "fetuses-at-risk approach," it is necessary to include the gestational age at the time the fetus died in utero. In singleton pregnancy, delivery is attempted promptly in the event of intrauterine fetal death. Therefore, a relatively accurate fetal survival time can be estimated. However, for multiple pregnancies, especially in the case of dichorionic twins, the standard management is continuing pregnancy until term period even if the one fetus dies in utero. Therefore, it is sometimes impossible to estimate the exact intrauterine survival time of dead fetus. From this point of view, we considered the "birth-based approach" better for this study.

This point was added to discussion along with the references suggested by the reviewers. (Lines: 274–292)

[Comment]

2. Maternal and child health indicators is a very broad term and as this manuscript focuses exclusively on perinatal mortality, the authors should consider using more specific terms in the manuscript title.

[Response]

Per the reviewer’s suggestion, we have changed the manuscript title to “Time trend analysis of perinatal mortality, stillbirth, and early neonatal mortality of multiple pregnancies for each gestational week from the year 2000 to 2019: A population-based study in Japan.”

[Comment]

3. The authors should include a brief description of the study population, state the statistical methods used to analyse the data, and present some results within the abstract. For example, where there any major exclusion criteria? What was the gestational age threshold for the database? The authors should also consider presenting overall mortality rates before describing trends over time.

[Response]

Following the reviewers' suggestions, we have added the brief methodology and results to the abstract. (Lines 21–37)

[Comment]

4. Lines 27-34 should be omitted. These are explanations which are more suitable for the discussion section than the abstract.

[Response]

As the reviewers suggested, we have deleted this part.

[Comment]

5. The second sentence (lines 42-46) should be revised to improve clarity.

6. Please also include references for lines 42-46

[Response]

Following the reviewers' suggestions, the description has been revised as follows, along with the corresponding references; 

In particular, complicated multiple pregnancies force obstetricians to decide the specific timing of delivery, with consideration of both maternal and fetal risks. Obstetricians are often required to predict the ideal timing of delivery, balancing between avoiding maternal complications that increase with later gestational age (e.g. gestational hypertension, preeclampsia [1,2], placental abruption [2], thromboembolism [2]) and preventing fetal immaturity due to preterm birth. Moreover, diseases specific to multiple pregnancies, such as twin-to-twin transfusion syndrome, usually worsen the prognosis of newborns. (Lines: 45–52)

[Comment]

7. For international audiences who are not familiar with the Ministry of Health, Labour and Welfare database, the authors should provide additional information about the database and the quality of the data collected. For example, what proportion of the birthing population in Japan is captured by the database? What are the criteria for inclusion in the database – what gestational age or birthweight thresholds are used? Are births with a congenital anomaly included or excluded? How valid and reliable are the data from the database? Are there any published validation studies using these data?

[Response]

In Japan, the "Family Register Act" stipulates that all births, deaths, and stillbirths that occur in Japan should be reported to the Ministry of Health, Labor and Welfare (MHLW). MHLW aggregates data based on the notification and publishes a database every year. The data collection rate is 100%, but it is not possible to include illegal births, stillbirths, or abortions. Live birth and stillbirth are grouped into one category for less than 20 gestational weeks and 49 weeks or more, respectively, and data for 20 to 48 weeks are published for each gestational week. Birth weights of less than 500 g and 6,900 g or more are grouped into one category, and data for birth weights 500 to 6,900 g are published every 100 g. Births with defects are also included. Since this database is used as a national official basis for policy making, it is considered to be the most reliable database for population studies in Japan. 

The study of multiple pregnancies using this database is taken as an example from JAMA 1996 by Minkami et al. (reference No.7) These points have been added to materials and methods, and the introduction. (Lines: 74–84)

[Comment]

8. The authors should state which statistical methods were used. Have the authors considered formal tests such as linear regression for the trends over time presented?

[Response]

We have added a "data analysis" subsection to the "Materials and Methods" section. (Lines: 117–125)

The raw data obtained were aggregated every 5 years (2000–2004, 2005–2009, 2010–2014, and 2015–2019). We analyzed the proportion of the number of births for each week of pregnancy to the total of each singleton and multiple pregnancy. For perinatal related health indicators (perinatal mortality, stillbirth, and neonatal mortality), the obtained data were plotted on graphs for each week of pregnancy without using a regression curve. Stillbirth weights were aggregated every 4 weeks with the exception of extreme-moderate preterm (22–33 weeks), late preterm (34–36 weeks), and term (after 37 weeks), each of which was aggregated as an individual category. This stillbirth weights were displayed as a histogram. 

[Comment]

9. In the text of the results, the authors should first describe the study population before referring to Table 1. For example, present the total number of births over the study period, describe any births excluded from the analyses for whatever reasons.

[Response]

In accordance with the reviewer’s suggestion, the following description has been added at the beginning of the Results section.

Between 2000 and 2019, there were 21,068,275 live births, 67,666 stillbirths, and 16,443 early neonatal deaths, excluding 7,148 (7,104 singletons, 44 multiple births) whose number of gestational weeks at birth was less than 22 weeks or unknown. (Lines: 135-137)

[Comment]

10. In Table 1, please include the number of stillbirths and neonatal deaths.

[Response]

As the reviewers pointed out, we have added stillbirths and neonatal deaths to the table.

[Comment]

11. Mortality rates should be presented as per 1000 rather than per 100 (%).

[Response]

Per the reviewer’s comment, the text and table description have been revised.

[Comment]

12. Lines 124-128 – Perhaps this is referring to the highest proportion of births occurring at 39 weeks for singletons and 37 weeks for multiples, rather than maximum number of gestational weeks?

[Response]

In accordance with the reviewer’s suggestion, the description in the text has been revised. (Lines: 119–120)

[Comment]

13. The study limitations should be moved so that they follow on from the discussion of study strengths.

[Response]

As the reviewer pointed out, the description of limitations have been moved. (Lines:232-239)

[Comment]

14. If the ‘fetus-at-risk’ approach is used, the discussion on intersecting mortality curves (lines 227-243) will likely need to be revised.

[Response]

In this study, we have used the "birth-based approach." Additionally, we have added a description of the rationale behind our choice. (Lines: 274–292)

The paradoxical mortality curves in this study can be explained by the point that this study was conducted using the "birth-based approach." The "fetuses-at-risk approach" is considered more appropriate as an epidemiological model of fetal mortality in a population with some maternal risk factors (e.g., smoking, hypertension). If multiple pregnancies are considered to be a risk factor for the fetuses, it is more appropriate to use the "fetuses-at-risk approach." However, in the case of multiple pregnancies, specific issues should be considered. In the case where one fetus died in utero, the gestational age at stillbirth of the deceased fetus is recorded as the gestational age at delivery with the other live infant. To use the reliable "fetuses-at-risk approach," it is necessary to include the gestational age at the time the fetus died in utero. In singleton pregnancy, delivery is attempted promptly in the event of intrauterine fetal death. Therefore, a relatively accurate fetal survival time can be estimated. However, for multiple pregnancies, especially in the case of dichorionic twins, the standard management is continuing pregnancy until term period even if the one fetus dies in utero. Therefore, it is sometimes impossible to estimate the exact intrauterine survival time of dead fetus. From this point of view, we considered the "birth-based approach" better for this study.

We would like to again thank you for your constructive comments, which have greatly enriched our manuscript.

---

## [Decision Letter · Decision Letter 1]

5 Jul 2022

PONE-D-21-26832R1Time trend analysis of perinatal mortality, stillbirth, and early neonatal mortality of multiple pregnancies for each gestational week from the year 2000 to 2019: A population-based study in JapanPLOS ONE

Dear Dr Francis Kiweewa,

Thank you for submitting your manuscript to PLOS ONE and for addressing the concerns raised by the reviewers. Before accepting the manuscript for publication, we invite you to address the following comments raised by the academic editor:==============================

ACADEMIC EDITOR: 

Dear authors, thanks for choosing PLOS ONE to publish this important work. I also thank you for addressing the comments of the reviewers. In order to improve readability and clarity of the manuscript and thereby its scientific and practical contribution to the filed, I invite you to address the following points.

The method section needs revision. Rationale for the selection of the analytical and methodological approaches should be described in the methods section. For instance, it would increase the merit of the article if you include brief introduction of birth-based approach and the "fetuses-at-risk approach” in the methods section. Note that, what has been included in the discussion is not enough and there should be some introduction of the method in the methods section. 

We look forward to receiving your revised manuscript.

Kind regards,

Garumma Tolu Feyissa, PhD

Academic Editor

PLOS ONE

Journal Requirements:

Reviewers' comments:

Reviewer's Responses to Questions

**Comments to the Author**

1. If the authors have adequately addressed your comments raised in a previous round of review and you feel that this manuscript is now acceptable for publication, you may indicate that here to bypass the “Comments to the Author” section, enter your conflict of interest statement in the “Confidential to Editor” section, and submit your "Accept" recommendation.

Reviewer #1: All comments have been addressed

Reviewer #2: All comments have been addressed

2. Is the manuscript technically sound, and do the data support the conclusions?

Reviewer #1: Yes

Reviewer #2: (No Response)

3. Has the statistical analysis been performed appropriately and rigorously? 

Reviewer #1: Yes

Reviewer #2: (No Response)

4. Have the authors made all data underlying the findings in their manuscript fully available?

Reviewer #1: Yes

Reviewer #2: (No Response)

5. Is the manuscript presented in an intelligible fashion and written in standard English?

Reviewer #1: Yes

Reviewer #2: (No Response)

6. Review Comments to the Author

Reviewer #1: Thanks for the opportunity of reviewing the revised version of this manuscript, all concerns have been addressed appropriately.

Reviewer #2: (No Response)

7. PLOS authors have the option to publish the peer review history of their article (what does this mean?). If published, this will include your full peer review and any attached files.

Reviewer #1: **Yes: **Kamran Hessami

Reviewer #2: No

---

## [Author Response · Author response to Decision Letter 1]

11 Jul 2022

Responses to Academic Editor:

The authors would like to thank the Academic Editor for the constructive critique to improve the manuscript. We have made every effort to address the issues raised and to respond to all comments. The revisions are indicated in red font in the revised manuscript. We hope that our revisions meet the Academic Editor’s expectations.

[Comment]

The method section needs revision. Rationale for the selection of the analytical and methodological approaches should be described in the methods section. For instance, it would increase the merit of the article if you include brief introduction of birth-based approach and the "fetuses-at-risk approach” in the methods section. Note that, what has been included in the discussion is not enough and there should be some introduction of the method in the methods section.

[Response]

We would like to thank the Editor for the comment. As the Editor suggested, we have moved the methodology description from the Discussion to the Methods section and compressed this description in the Discussion section. The revised parts are as follows:

“These indicators were calculated in the "birth-based approach" [10]. The "fetuses-at-risk approach" is considered more appropriate as an epidemiological model of fetal mortality in a population with some maternal risk factors (e.g., smoking and hypertension). If multiple pregnancies are considered to be a risk factor for the fetuses, it is more appropriate to use the "fetuses-at-risk approach." However, in multiple pregnancies, specific issues should be considered. In cases where a fetus dies in the uterus, the gestational age at stillbirth of the deceased fetus is recorded as that at delivery of the other live infant. To use the "fetuses-at-risk approach," it is necessary to include the gestational age at the time when the fetus died in the uterus. However, especially in cases of dichorionic twins, the standard management in twins is continuing pregnancy until the term period even if one fetus dies in the uterus. Therefore, it is usually impossible to estimate the exact intrauterine survival time of the dead fetus from the database. From these points of view, we thought that it would be better to apply the "birth-based approach" at this stage.” (Lines 114–127)

“One reason for these paradoxical mortality curves (Figs 2–4) in this study may be the fact that this study was conducted using the "birth-based approach" [10]. It has been argued this paradoxical appearance of gestational age-specific perinatal mortality curves in twins vs. singletons [22–24]. Yudkin et al. tried to explain this paradox and proposed the "fetuses-at-risk approach" formulation [25]. To use the aforementioned approach, it is necessary to include the gestational age at the time when the fetus died in the uterus. If the estimated period of death of the deceased twin fetus can be tracked on the database, it will be possible to apply the "fetuses-at-risk approach" to verify this paradox.” (Lines 285–293)

We would like to thank again the Editor for the constructive comments, which have helped us significantly improve the quality of our work.

---

## [Editor Report · Decision Letter 2]

13 Jul 2022

Time trend analysis of perinatal mortality, stillbirth, and early neonatal mortality of multiple pregnancies for each gestational week from the year 2000 to 2019: A population-based study in Japan

PONE-D-21-26832R2

Dear Dr. Nasahiko Nakata,

We’re pleased to inform you that your manuscript has been judged scientifically suitable for publication and will be formally accepted for publication once it meets all outstanding technical requirements.

Kind regards,

Garumma Tolu Feyissa, PhD

Academic Editor

PLOS ONE
---

## [Editor Report · Acceptance letter]

15 Jul 2022

PONE-D-21-26832R2 

Time trend analysis of perinatal mortality, stillbirth, and early neonatal mortality of multiple pregnancies for each gestational week from the year 2000 to 2019: A population-based study in Japan 

Dear Dr. Nakata:

I'm pleased to inform you that your manuscript has been deemed suitable for publication in PLOS ONE. Congratulations! Your manuscript is now with our production department. 

Kind regards, 

on behalf of

Dr. Garumma Tolu Feyissa 

Academic Editor

PLOS ONE